# A STUDY OF UNSUPERVISED EVALUATION METRICS FOR PRACTICAL AND AUTOMATIC DOMAIN ADAPTATION

## ABSTRACT

Unsupervised domain adaptation (UDA) methods facilitate the transfer of models to target domains without labels. However, these methods necessitate a labeled target validation set for hyper-parameter tuning and model selection. In this paper, we aim to find an evaluation metric capable of assessing the quality of a transferred model without access to target validation labels. We begin with the metric based on mutual information of the model prediction. Through empirical analysis, we identify three prevalent issues with this metric: 1) It does not account for the source structure. 2) It can be easily attacked. 3) It fails to detect negative transfer caused by the over-alignment of source and target features. To address the first two issues, we incorporate source accuracy into the metric and employ a new MLP classifier that is held out during training, significantly improving the result. To tackle the final issue, we integrate this enhanced metric with data augmentation, resulting in a novel unsupervised UDA metric called the Augmentation Consistency Metric (ACM). Additionally, we empirically demonstrate the shortcomings of previous experiment settings and conduct large-scale experiments to validate the effectiveness of our proposed metric. Furthermore, we leverage our metric to automatically search for the optimal set of hyper-parameters, achieving superior performance comparable to manually tuned sets across four common benchmarks.

## 1 INTRODUCTION

Deep neural networks, when trained on extensive datasets, have demonstrated exceptional performance across various computer vision tasks such as classification (Liu et al., 2022; Radford et al., 2021), object detection (Carion et al., 2020; Zhang et al., 2022), and semantic segmentation (Chen et al., 2018; Xie et al., 2021). Some even exhibit remarkable generalization to unseen domains (Gu et al., 2021; Zou et al., 2023). But performance in specific domains can always be enhanced through fine-tuning with labels. The challenge arises in real-world applications where manually labeling ample data for fine-tuning is both costly and impractical. Consider a household robot equipped with a vision system trained on vast vision datasets (She et al., 2020). When introduced to a new home, the robot is anticipated to automatically adapt to this new environment by collecting images from the house. Yet, expecting homeowners to label these images is both burdensome and unrealistic.

Unsupervised domain adaptation (UDA) emerged as a solution to this problem. Over recent years, a plethora of UDA methods (Long et al., 2018; Saito et al., 2018; Zhang et al., 2019; Jin et al., 2020; Tanwisuth et al., 2021) have been developed to facilitate transfer to label-free target domains. While accuracy in these domains has seen improvement, it often hinges on meticulous tuning using a labeled validation set from the target domain, which can be resource-intensive. Referring back to the household robot example, creating validation sets would necessitate precise labels from homeowners, which hinders the fully automatic adaptation. One could pre-test UDA method hyper-parameters in sample homes before actual deployment, selecting parameters that perform well across these homes. However, as UDA methods tend to be hyper-parameter sensitive, different domains might demand distinct hyper-parameter configurations. It is challenging to finalize an optimal set before deployment.

DEV (You et al., 2019) first proposed a general UDA evaluation metric, which uses the importance-weighted validation method (Sugiyama et al., 2007) with a variance control term. Later, SND (Saito et al., 2021) suggested that a good transfer model should have a compact neighborhood for each target feature and introduced the soft neighborhood density metric. However, upon more comprehensive

| Metrics | Ar $\rightarrow$ Cl | Ar $\rightarrow$ Pr | Ar $\rightarrow$ Rw | Cl $\rightarrow$ Ar | Cl $\rightarrow$ Pr | Cl $\rightarrow$ Rw | Pr $\rightarrow$ Ar | Pr $\rightarrow$ Cl | Pr $\rightarrow$ Rw | Rw $\rightarrow$ Ar | Rw $\rightarrow$ Cl | Rw $\rightarrow$ Pr | Avg |
|---|---|---|---|---|---|---|---|---|---|---|---|---|---|
| DEV (You et al., 2019) | 4.50 | 2.62 | 1.98 | 1.92 | 0.53 | 9.25 | 1.78 | 16.9 | 8.10 | 0.82 | 0.61 | **0.00** | 4.07 |
| SND (Saito et al., 2021) | 16.6 | 2.62 | 1.91 | 22.6 | 24.4 | 19.3 | 14.1 | 16.9 | **0.00** | 0.68 | 16.9 | 0.37 | 11.4 |
| ACM (ours) | **1.53** | **0.00** | **0.00** | **0.68** | **0.00** | **1.07** | **0.00** | **0.00** | **0.00** | **0.00** | **0.00** | **0.00** | **0.67** |

Table 1: In all 12 transfer tasks from the OfficeHome dataset, we employ CDAN (Long et al., 2018) as the training method. The hyper-parameter space is defined as {trade-off={0.1,0.2,0.3,0.5,1.0,2.0,3.0}}. We show the deviations between the optimal model determined by metrics and the true best target accuracy. In SND paper (Saito et al., 2021) they only presented results for Ar $\rightarrow$ Pr and Rw $\rightarrow$ Ar.

and detailed experiments with UDA evaluation metrics, we discovered previous evaluation metrics often failed to select suitable models in most scenarios, as shown in Tab 1. This is because their metrics are based on assumptions that do not always hold true in a wide range of scenarios. For more related discussions, refer to Related Works.

This realization led us to rethink and reassess what a robust UDA evaluation metric should be like. A robust UDA evaluation metric, as we define, should satisfy three principles: 1) Target Unsupervised. 2) Consistency with target accuracy in a wide range of scenarios. 3) Robustness: the metric should not be vulnerable to deliberately designed training methods and hyper-parameter sets. Previous works generally assume the first two principles; we augment this understanding by introducing the "Robustness" principle. This new perspective, inspired by the study of adversarial attacks in neural networks (Goodfellow et al., 2014b), emphasizes the importance of designing metrics resistant to potential failure cases, thus leading to more robust evaluations.

Building upon this redefined understanding of a robust UDA evaluation metric, we turn our attention to a classic UDA algorithm, mutual information (Morerio et al., 2017; Shi & Sha, 2012), used as an evaluation metric that measures the confidence (entropy) and diversity of the model on target samples. We meticulously dissect the metric to evaluate its adherence to the aforementioned three principles. Our investigation reveals three significant drawbacks with the metric: 1) unaware of the alignment between the prediction and the label, 2) easily attacked by designed training methods, 3) cannot detect negative transfer caused by the over-alignment between the source and target features. To address these issues, we first incorporate source accuracy into the metric to retain the source label structure. Then, we employ an additional MLP classifier held out during training to defend against attacks. We refer to this new metric as Inception Score Metric for UDA (ISM). Finally, we integrate the metric with data augmentation and propose Augmentation Consistency Metric (ACM) to evaluate models beyond features-level consideration.

We also establish new experimental settings for validating evaluation metrics, which contain sufficient datasets, training methods, and hyperparameter sets. In large-scale validation experiments, our evaluation metrics demonstrate high consistency with target accuracy in most cases. The study of evaluation metrics also has the potential to advance AutoML (Zoph & Le, 2016; Pham et al., 2018) research in the context of UDA. We employ simple hyperparameter optimization (Akiba et al., 2019) to illustrate this concept. Experiments show that the hyper-parameters automatically discovered by our metrics outperform manually tuned hyper-parameters for four popular UDA methods.

## 2 RELATED WORKS

### 2.1 UNSUPERVISED DOMAIN ADAPTATION

Unsupervised domain adaptation (UDA) (Long et al., 2015) has been developed to save annotation effort during the transfer from the source domain to the target domain. Most UDA methods aim to reduce the divergence (Ben-David et al., 2006; 2010) between the source and target domains, e.g., Discrepancy-based UDA (Kang et al., 2019; Sun & Saenko, 2016), Domain Adversarial UDA (Ganin et al., 2016; Long et al., 2018; Saito et al., 2018; Zhang et al., 2019), self-supervised-based UDA (French et al., 2017; Jin et al., 2020). However, none has formally studied whether their methods can decide the best model or how to tune hyper-parameters without target labels.

### 2.2 MODEL SELECTION FOR UDA

Some previous papers (You et al., 2019; Saito et al., 2021) are also interested in the unsupervised evaluation metric for UDA, also known as the model selection for UDA.

**Importance Weighted Validation:** In (Long et al., 2018), they tune the trade-off parameter using importance-weighted cross-validation (Sugiyama et al., 2007). In the later work (You et al., 2019), Deep Embedded Validation (DEV) is proposed to select models based on importance weights and control variates. However, their DEV metric requires overlap between the support sets of two domain distributions. In practice, it could easily collapse when no overlap exists between two domain distributions or the source error becomes 0.

**Entropy-based Metric:** Morerio et al. (Morerio et al., 2017) used the predicted entropy of the target domain samples as the metric to tune the hyper-parameter. Although it is simple and convenient, it was pointed out that it cannot deal with blind confidence (Saito et al., 2021). SND (Saito et al., 2021) proposes to use the density of the target domain sample domain as an evaluation metric to solve this problem. However, SND cannot solve the "mode collapse" problem where all target samples are mapped into one feature point.

**Other Metrics:** BPDA (Zellinger et al., 2021) introduces a principle to balance the source supervised error and domain distance in a target error bound. However, their approach is limited to adjusting the trade-off hyper-parameter, leaving other hyper-parameters untouched. More recently, Dinu et al. (Dinu et al., 2023) propose linear aggregations of vector-valued models to ensemble various models trained under different hyper-parameters. However, their resultant model aggregates all models across diverse hyper-parameters, demanding significantly more computational resources than a singular model, which is not practical for vision tasks.

## 3 METHODS

### 3.1 PROBLEM DEFINITION

During the training of unsupervised domain adaptation, we have a labeled dataset sampled from the source domain, $\{(\boldsymbol{x}_i^s, y_i^s)\}_{i=1}^{n_s} \sim \mathcal{D}_s$ and an unlabeled dataset sampled from the target domain, $\{\boldsymbol{x}_j^t\}_{j=1}^{n_t} \sim \mathcal{D}_t$. We can apply off-the-shelf UDA methods (Ganin et al., 2016; Long et al., 2018; Zhang et al., 2019; Jin et al., 2020) to train a model $\boldsymbol{M}$ on these two training datasets. During the evaluation of UDA, we are given a labeled dataset from the source domain and an unlabeled dataset from the target domain, $\{(\tilde{\boldsymbol{x}}_i^s, \tilde{y}_i^s)\}_{i=1}^{\tilde{n}_s} \sim \mathcal{D}_s$ and $\{\tilde{\boldsymbol{x}}_j^t\}_{j=1}^{\tilde{n}_t} \sim \mathcal{D}_t$, which contain different samples from the training sets. Given a trained model $\boldsymbol{M}$, an unsupervised evaluation metric for UDA should compute a score to reflect the classification accuracy of the model on the target domain $\mathcal{D}_t$, based on the evaluation sets and the model $\boldsymbol{M}$. As a common practice, the model is decomposed into a feature generator $\mathbf{g}(\cdot)$ and a linear classifier $\mathbf{f}(\cdot)$: $\boldsymbol{M} = \mathbf{f}(\mathbf{g}(\cdot))$.

### 3.2 PRINCIPLES OF A ROBUST METRIC

We define the principles of a robust unsupervised evaluation metric for UDA as the following:

**1) Target Unsupervised:** The metric can only access the evaluation sets of UDA, $\{(\tilde{\boldsymbol{x}}_i^s, \tilde{y}_i^s)\}_{i=1}^{\tilde{n}_s}$ and $\{\tilde{\boldsymbol{x}}_j^t\}_{j=1}^{\tilde{n}_t}$, and the model $\boldsymbol{M}$. For versatility, the metric should be irrelevant to the training method.

**2) Consistency:** Given a bunch of models $\{\boldsymbol{M}_l\}_{l=1}^{n_m}$ trained with different UDA methods and different hyper-parameters, the metric score $\{\boldsymbol{S}_l\}_{l=1}^{n_m}$ should be consistent with the target classification accuracy $\{\boldsymbol{A}_l\}_{l=1}^{n_m}$ and this consistency holds for multiple UDA datasets.

**3) Robustness:** The metric should maintain consistency when we deliberately design the training method and the hyper-parameter to attack the metric. Typically, if the metric can be transformed to a training loss for UDA, the metric score should still be consistent with the target accuracy when training with this loss.

Our intuition is that a robust metric should reflect the target domain accuracy under various conditions. At the same time, a robust metric should not be vulnerable to attack, which avoids some methods of deliberately optimizing the metric and finding metric preferences. Just as the robustness of the neural network can be improved through the attack on the neural network (Goodfellow et al., 2014a), the analysis of the attack on the evaluation metric can help us construct a more robust evaluation metric.

| Metric | CDAN | MCC |
|---|---|---|
| Source Acc. | 10.7 | 5.2 |
| Entropy | 3.9 | 26.2 |
| MI | 4.4 | 14.4 |
| MI w. source | **0.3** | **0.5** |

Table 2: Using CDAN (Long et al., 2018) or MCC (Jin et al., 2020) as the training method, when search trade-off from {0.1, 0.2, 0.3, 0.5, 1.0, 2.0, 3.0, 5.0, 10.0}, "dev" of metrics. The results are averaged across 12 transfers in OfficeHome.

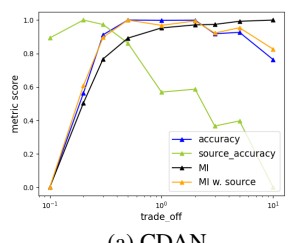
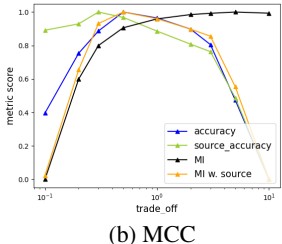

(a) CDAN      (b) MCC

Figure 1: On OfficeHome, using CDAN (Long et al., 2018) or MCC (Jin et al., 2020) as the training method, when the trade-off changes, the curves of various metrics. For display convenience, we normalize each metric to [0,1].

We utilize two measurements to measure the degree of consistency between the metric score and target accuracy:

**Pearson's correlation coefficient:**

$$corr(\{\boldsymbol{S}_l\}_{l=1}^{n_m}, \{\boldsymbol{A}_l\}_{l=1}^{n_m}) = \frac{\mathbb{E}(\boldsymbol{SA}) - \mathbb{E}(\boldsymbol{S})\mathbb{E}(\boldsymbol{A})}{\sigma_{\boldsymbol{S}}\sigma_{\boldsymbol{A}}}, \tag{1}$$

where $\sigma$ is the standard deviation.

**The deviation of Best Model**

$$dev(\{\boldsymbol{S}_l\}_{l=1}^{n_m}, \{\boldsymbol{A}_l\}_{l=1}^{n_m}) = \max_l \boldsymbol{A}_l - \boldsymbol{A}_{l^*}, \tag{2}$$

where $l^* = \operatorname{argmax}_l \boldsymbol{S}_l$ denotes the best model according to the metric. The metric with a higher correlation and lower deviation is more consistent with target accuracy.

### 3.3 Derivation of Our Metrics

#### 3.3.1 Combine with Source Accuracy

Originating from Semi-supervised learning (Grandvalet & Bengio, 2004), the Entropy of the prediction is commonly used in UDA methods (Vu et al., 2019) as a regularizer for unlabeled samples. Entropy can also serve as the evaluation metric for UDA (Morerio et al., 2017). As the Entropy metric is unaware of the "mode collapse" phenomenon, Mutual Information (Shi & Sha, 2012) adds the diversity term. The Mutual Information metric (MI) is defined as: $\boldsymbol{MI} = H(\mathbb{E}_{\tilde{\boldsymbol{x}}^t}[\boldsymbol{p}^t]) - \mathbb{E}_{\tilde{\boldsymbol{x}}^t}[H(\boldsymbol{p}^t)]$, where $\boldsymbol{p}^t$ denotes the prediction of the model on $\tilde{\boldsymbol{x}}^t$. MI only considers the quality of the target samples. While the source label information and the quality of the source feature are ignored. In some situations, the prediction of the target sample is not aligned with our desired label space. To avoid this problem, SND (Saito et al., 2021) suggests monitoring the source supervising loss or setting a threshold for source accuracy. However, we find this rough approach cannot help to determine the best model. To show the importance of source accuracy during evaluating UDA, we use CDAN (Long et al., 2018) or MCC (Jin et al., 2020) method to train the models with multiple trade-off hyper-parameters. We use the metric to evaluate the trained models and determine the best trade-off. As shown in Tab. 2 and Fig. 1, the Entropy metric and MI increase as the trade-off increases, but the target accuracy first increases and then decreases as the trade-off increases. We propose to combine MI and source accuracy directly. We first normalize MI into [0, 1], then add it with the source accuracy, as follows:

$$\boldsymbol{MI}_{w.source} = \mathbb{E}_{(\tilde{\boldsymbol{x}}^s, \tilde{y}^s)} I[\operatorname{argmax}_k[\boldsymbol{p}^s] = \tilde{y}^s] + \frac{\boldsymbol{MI}}{2\log K} + \frac{1}{2}, \tag{3}$$

where $\boldsymbol{p}^s$ denotes the prediction of the model on $\tilde{\boldsymbol{x}}^s$, and $K$ is the number of classes. As shown in Tab. 2 and Fig. 1, this simple combination can balance MI on the target domain and source accuracy. If we view the MI term as the similarity between two domains, this metric formally follows Ben David's theory (Ben-David et al., 2006), where the source error and the domain discrepancy bound the target error.

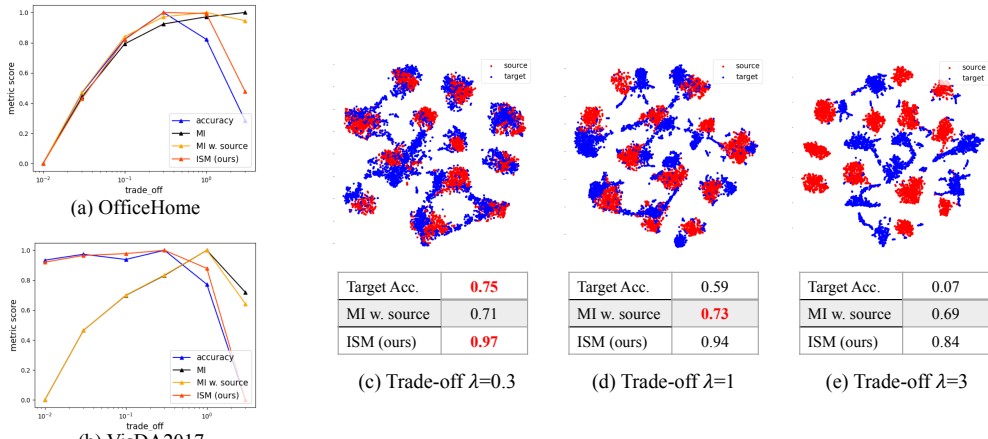

Figure 2: On OfficeHome and VisDA2017, use Mutual Information as the training algorithm. (a)-(b) When the trade-off value changes, the Mutual Information, ISM metric, and target domain accuracy change. (c)-(e) The tSNE (van der Maaten & Hinton, 2008) visualization of model features and each metric scores when the trade-offs are equal to 0.3, 1, and 3 on VisDA2017.

### 3.3.2 ROBUSTNESS PROPERTY

As mentioned in Section 3.2, a robust metric should not be vulnerable when we maliciously increase the metric score. To attack $MI_{w.source}$ metric, we train the model with the following loss:

$$Loss = \mathbb{E}_{(\boldsymbol{x}^s, y^s)}[-\log \boldsymbol{p}_{y^s}] + \lambda(\sum_k \hat{\boldsymbol{p}}_k \log \hat{\boldsymbol{p}}_k - \mathbb{E}_{\boldsymbol{x}^t}[\sum_k \boldsymbol{p}_k \log \boldsymbol{p}_k]), \qquad (4)$$

where $\hat{\boldsymbol{p}}_k$ is the average prediction for class $k$ within a batch. This loss is actually the UDA method used in (Shi & Sha, 2012), which maximizes mutual information. We use this loss to train the models with different trade-off hyper-parameters $\lambda$. As shown in Fig. 2, MI and MI w.source would prefer a large trade-off, but target accuracy decreases dramatically as the trade-off becomes larger. To investigate the cause, we use tSNE (van der Maaten & Hinton, 2008) to visualize the source and target features. As demonstrated in Fig. 2 (c)-(e), source features (red) form clear clusters, but target features are pushed away as the trade-off increases. The MI metric is almost unaware of this phenomenon because the training loss is finding the preference of MI.

The MI metric is vulnerable for two reasons: the evaluation metric is fully exposed to the training process, and the linear classifier **f** cannot detect feature outliers. To solve this problem, we propose to train a new two-layer MLP on top of the source evaluation feature. Then we use the Mutual Information of this MLP classifier on the target evaluation feature as the evaluation metric. The new metric $ISM$ can be formalized as follows:

$$\boldsymbol{IS} = H(\mathbb{E}_{\tilde{\boldsymbol{x}}^t}[\boldsymbol{q}^t]) - \mathbb{E}_{\tilde{\boldsymbol{x}}^t}[H(\boldsymbol{q}^t)], \qquad (5)$$

$$\boldsymbol{ISM} = \mathbb{E}_{(\tilde{\boldsymbol{x}}^s, \tilde{y}^s)} I[\underset{k}{\operatorname{argmax}}[\boldsymbol{p}^s] = \tilde{y}^s] + \frac{\boldsymbol{IS}}{2\log K} + \frac{1}{2}, \qquad (6)$$

where $\boldsymbol{q}^t = \mathbf{h}(\mathbf{g}(\tilde{\boldsymbol{x}}^t))$ and **h** is the MLP classifier trained on the source evaluation feature. Noticeably, we still use the classifier of the model to compute the source accuracy, which evaluates whether the classifier **f** is well-trained. As **h** is held out during the UDA training and the two-layer MLP is more expressive, $ISM$ is not vulnerable to attack. As shown in Fig. 2, when trained with the mutual information loss, $ISM$ can be well consistent with the target accuracy. $ISM$ is short for the Inception Score Metric for UDA because it is formally similar to the Inception Score for generative models (Salimans et al., 2016). The inception score for the generative model utilizes the mutual information of the ImageNet pretrained inception network (Szegedy et al., 2016). Instead, we train an MLP classifier based on the supervision of the source evaluation set and combine it with the original source accuracy.

### 3.3.3 INPUT-LEVEL CONSISTENCY

In the experiments, our ISM already shows surprising consistency with target accuracy. However, we find that in some situations where we use feature alignment-based UDA methods, e.g., DANN and

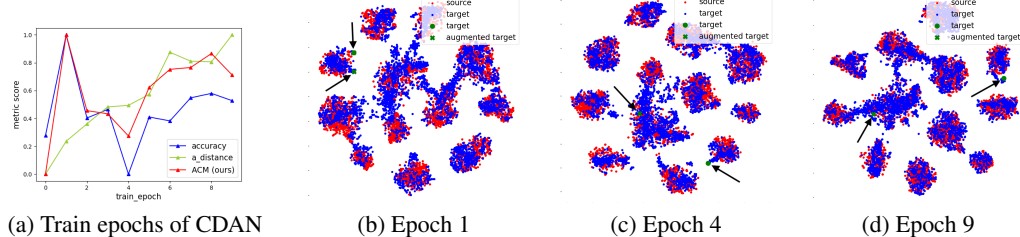

| (a) Train epochs of CDAN | (b) Epoch 1 | (c) Epoch 4 | (d) Epoch 9 |

Figure 3: On VisDA2017, using CDAN as the training method. (a) As training epochs grow, the curves of target accuracy, A-distance Metric, and ACM (ours). For display convenience, we normalize each metric to [0,1]. (b)-(d) The tSNE visualization of model features and a target feature (green circle) that gradually misclassified paired with the feature of its augmented view (green cross).

CDAN, the target accuracy might decrease with the training process. In these situations, the source and target features are well aligned, and source accuracy does not degrade, but target accuracy does not improve. We interpret this phenomenon as some target features being pulled to the wrong source class to make the target feature distribution the same as the source. We show this phenomenon in Fig 3: the target accuracy approaches the maximum in the first epoch while the feature distributions of the two domains are not aligned. This phenomenon violates the common UDA assumption that samples embedded nearby are likely to share the labels (Saito et al., 2021). In these situations, the metrics that consider the features and predictions of two domains are hard to determine the model.

To solve this problem, we propose to use the consistency between the target sample and its augmented view to detect whether target features are over-aligned. This is based on the finding that for misaligned target samples, the features are more unstable to data augmentation. In Fig 3, a target feature (green dot) is close to its data-augmented feature (green cross) at epoch 1, but the two become farther away as it is over-aligned. We define our Augment Consistency Metric (ACM) as follows:

$$AC = \mathbb{E}_{\tilde{\boldsymbol{x}}^t} I[\underset{k}{\arg\max}[\boldsymbol{q}^t] = \underset{k}{\arg\max}[\boldsymbol{q}^{t\prime}]] \tag{7}$$

$$ACM = \mathbb{E}_{(\tilde{\boldsymbol{x}}^s,\tilde{y}^s)} I[\underset{k}{\arg\max}[\boldsymbol{p}^s] = \tilde{y}^s] + \frac{1}{2}(AC + \frac{H(\mathbb{E}_{\tilde{\boldsymbol{x}}^t}[\boldsymbol{q}^t])}{\log K}), \tag{8}$$

where $\boldsymbol{q}^{t\prime} = \mathbf{h}(\mathbf{g}(\tilde{\boldsymbol{x}}^{t\prime}))$ denotes the prediction of the MLP classifier on the data-augmented sample $\tilde{\boldsymbol{x}}^{t\prime}$. We use the MLP prediction instead of the original classifier to make it robust and combine it with the diversity term to avoid "mode collapse". As shown in Fig 3, after taking input-level disturbance into consideration, ACM is consistent with target accuracy in the over-alignment situation. While input-level consistency has been utilized as a UDA method (French et al., 2017), we are the first to study it as an evaluation metric.

### 3.4 FLAWS IN PREVIOUS METRICS

In this section, we reveal the flaws in the experiment settings and metrics of previous works (You et al., 2019; Saito et al., 2021). It is important to experiment with sufficient datasets, training methods, and hyper-parameter sets to verify a robust evaluation metric for UDA. Based on the findings, we will construct our experiment settings in the experiment section 4.1.

**More Datasets and Training Methods are Important.** In the DEV (You et al., 2019) and SND (Saito et al., 2021) papers, they only used part of UDA datasets and part of UDA training methods, such as in the DEV paper (You et al., 2019), only the CDAN training method is used on Office31, and only the MCD training method is used on VisDA. In the SND paper (Saito et al., 2021), although they used four training algorithms, they only test metrics on two transfer tasks of OfficeHome, Ar → Pr and Rw → Ar, and one transfer task of DomainNet, real → clipart. However, **it is easy to draw wrong conclusions by testing evaluation metrics on the part of datasets.** As shown in the table 1, although DEV and SND perform well on some transfer tasks, e.g., Ar → Pr and Rw → Ar, they perform poorly on most transfer tasks. This is likely because the Ar → Pr and Rw → Ar transfer tasks are closer to the assumptions of their metric, e.g., SND assumes that tighter intra-class domains have higher accuracy. Our evaluation metric ACM achieves excellent results on all 12 transfer tasks.

It is also important to validate the evaluation metrics using multiple training methods on each dataset. In the experiment, we employ five classic UDA algorithms. Additionally, we will investigate an

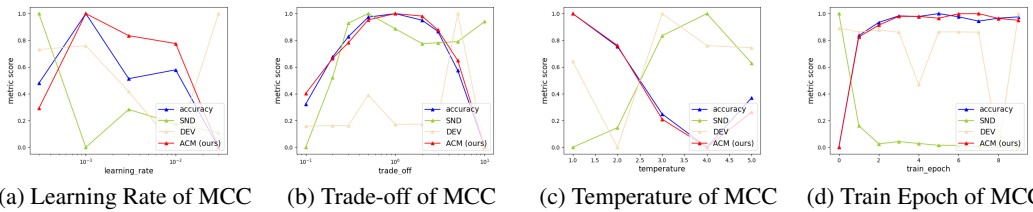

| (a) Learning Rate of MCC | (b) Trade-off of MCC | (c) Temperature of MCC | (d) Train Epoch of MCC |

Figure 4: On OfficeHome, when independently changing different hyper-parameters of MCC, curves of various evaluation metrics (averaged across 12 transfer tasks). We normalize each metric to [0,1].

| Training Method | Sparse hyper-parameter space | Dense hyper-parameter space |
|---|---|---|
| Source only | {lr={1e-3,1e-2}, wd={1e-4,1e-3}, train-epoch={1...10}} | - |
| DANN or CDAN | {lr={1e-3,1e-2}, wd={1e-4,1e-3}, lr-multi-D={0.1 ,1,10}, trade-off={0.1,1,10} bottleneck-dim= {256,512}, train-epoch={1...10} } | {lr={3e-4,1e-3,3e-3,1e-2,3e-2}, wd={1e-4,3e- 4,1e-3}, lr-multi-D={0.1,0.3,1,3,10}, trade-off={0.1,0.3,1,3,10}, bottleneck-dim= {256,512}, train-epoch={1...10}} |
| MCC | {lr={1e-3,1e-2}, wd={1e-4,1e-3}, trade-off={0.1,1,10}, train-epoch={1...10}} | {lr={3e-4,1e-3, 3e-3,1e-2,3e-2}, wd={1e-4,3e-4,1e-3 }, temperature={1,2,3,4,5}, trade-off={0.1,0.3,1,3,10}, bottleneck-dim={512,1024,2048 }, train-epoch={1...10}} |
| MDD | {lr={4e-4,4e-3}, wd={5e-5,5e-4}, trade-off={0.1,1,10 }, train-epoch={1...10}} | {lr={3e-4,1e-3,3e-3,1e-2,3e-2}, wd={1e-4,3e-4,1e-3 }, margin={1,2,3,4,5}, trade-off={0.1,0.3,1,3,10}, bottleneck-dim={512,1024,2048 }, train-epoch={1...10}} |

Table 3: The hyper-parameter spaces used in the experiments. In the sparse hyper-parameter space, we perform grid search over all combinations of hyper-parameters to analyze the consistency. In the dense hyper-parameter space, we use our metric to search for optimal hyper-parameters.

interesting question: Can the metrics be used to select training methods for a transfer scenario? This problem is very important in practice because a large number of UDA algorithms have been proposed, so for a transfer scenario, it is troublesome to choose the most suitable training method.

**Larger and Wider Hyper-parameter Sets are Important.**

For different datasets, the optimal hyper-parameters may vary by ten times (Jiang et al., 2020). In addition, it is often necessary to tune multiple hyper-parameters of the method to achieve the optimal result. However, in the previous DEV and SND papers, only one hyper-parameter was adjusted for each training method, and the adjustment range of hyper-parameters was small. As shown in Fig. 4, when the hyper-parameter of MCC changes, previous metrics DEV and SND cannot be consistent with target accuracy. In addition, we also found that a larger selection interval will lead to different conclusions from the small ones. For example, when the trade-off values are changed, SND can maintain the same accuracy rate between 0.1 and 1.0, but if the trade-off value increases to 10.0, SND will give the opposite result. Finally, it can be seen from the figure that our ACM always maintains high consistency with target accuracy and can select the optimal value for various hyper-parameters.

## 4 EXPERIMENTS

### 4.1 EXPERIMENTAL SETTINGS

**Datasets.** UDA datasets studied in the main paper: 1) *OfficeHome* (Venkateswara et al., 2017) consists of 15,500 images with 65 classes from four domains: Artistic images (Ar), Clip art (Cl), Product images (Pr), and Real-world (Rw). There are 12 transfer tasks among these domains. 2) *VisDA2017* (Peng et al., 2017) contains 12 categories and over 280,000 images from the Synthetic source domain and Real-world target domain. 3) *DomainNet* (Peng et al., 2019) is a large-scale dataset for domain adaptation and contains 345 categories from six domains. We select four domains for our experiments: Clipart (c), Painting (p), Real (r), and Sketch (s). We only study single-source domain adaptation of DomainNet. There are 12 transfer tasks among these domains. 4) *Office31* (Saenko & Kulis, 2010) is a relatively small dataset containing 4652 images with 31 categories from three domains. Results on Office31 are provided in the Appendix.

**Training Methods.** We use five popular UDA methods to get trained models. **1) Source only 2) DANN** (Ganin et al., 2016) **3) CDAN** (Long et al., 2018) **4) MDD** (Zhang et al., 2019) **5) MCC** (Jin

| Training Method | Source only | | DANN | | CDAN | | MDD | | MCC | | ALL | |
|---|---|---|---|---|---|---|---|---|---|---|---|---|
| Metric | corr | dev | corr | dev | corr | dev | corr | dev | corr | dev | corr | dev |
| $\mathcal{A}$-distance | -0.81 | 6.99 | 0.55 | 6.47 | -0.17 | 6.56 | 0.58 | 2.25 | 0.37 | 1.66 | 0.44 | 7.76 |
| MCD | -0.58 | 8.03 | 0.77 | 6.29 | -0.26 | 4.73 | 0.86 | 0.57 | -0.06 | 66.29 | 0.5 | 8.67 |
| DEV | 0.12 | 7.39 | -0.08 | 4.37 | -0.08 | 4.71 | -0.09 | 47.54 | -0.11 | 64.27 | -0.03 | 64.27 |
| Entropy | -0.14 | 8.99 | 0.56 | 4.81 | -0.29 | 9.34 | 0.64 | 1.17 | -0.06 | 66.29 | -0.34 | 66.29 |
| SND | -0.74 | 8.12 | 0.46 | 8.51 | -0.72 | 9.81 | -0.55 | 50.33 | -0.58 | 67.65 | -0.42 | 52.12 |
| MI | 0.06 | 6.26 | 0.58 | 3.92 | -0.07 | 3.72 | 0.81 | 0.0 | 0.03 | 5.4 | 0.45 | 5.4 |
| ISM | 0.84 | 0.31 | 0.75 | 3.92 | 0.42 | 1.23 | 0.75 | 0.40 | 0.88 | 0.66 | 0.59 | 1.66 |
| ACM | 0.80 | 2.38 | 0.79 | 1.18 | 0.61 | 0.98 | 0.85 | 0.0 | 0.93 | 1.66 | 0.76 | 1.66 |

Table 4: Consistency between metrics of UDA and target accuracy on VisDA2017, when models are trained by different UDA methods and hyper-parameters. The "ALL" method denotes assembling models trained by all five methods. The higher the Pearson's correlation ("corr") and the lower the deviation ("dev"), the better the metric. **Red score** is the best and blue score is the second best.

| Training Method | Source only | | DANN | | CDAN | | MDD | | MCC | | ALL | |
|---|---|---|---|---|---|---|---|---|---|---|---|---|
| Metric | corr | dev | corr | dev | corr | dev | corr | dev | corr | dev | corr | dev |
| $\mathcal{A}$-distance | 0.32 | 5.8 | 0.71 | 1.5 | 0.67 | 1.82 | 0.93 | 1.27 | 0.45 | 8.62 | 0.56 | 6.71 |
| MCD | 0.57 | 1.12 | 0.75 | 2.55 | 0.69 | 1.79 | 0.93 | 1.13 | 0.76 | 1.03 | 0.71 | 3.16 |
| DEV | 0.01 | 4.32 | 0.06 | 8.14 | 0.06 | 3.4 | 0.11 | 9.54 | -0.02 | 11.51 | 0.01 | 12.42 |
| Entropy | -0.64 | 8.20 | 0.43 | 4.28 | 0.88 | 1.56 | 0.88 | 1.98 | 0.38 | 24.43 | 0.52 | 24.52 |
| SND | -0.60 | 8.17 | -0.23 | 7.90 | 0.07 | 9.24 | -0.90 | 54.57 | -0.20 | 12.40 | -0.25 | 34.75 |
| MI | -0.60 | 6.78 | 0.45 | 4.25 | 0.88 | 1.40 | 0.91 | 1.98 | 0.37 | 22.83 | 0.52 | 22.93 |
| ISM | 0.72 | 1.47 | 0.6 | 1.68 | 0.91 | 1.15 | 0.97 | 1.45 | 0.70 | 1.96 | 0.88 | 1.96 |
| ACM | 0.75 | 1.37 | 0.77 | 1.16 | 0.90 | 1.13 | 0.95 | 0.93 | 0.94 | 1.36 | 0.93 | 1.73 |

Table 5: The "corr" and "dev" results are averaged over the 12 transfer tasks of OfficeHome.

et al., 2020). The implementations of these methods all follow TL-Lib (Jiang et al., 2020). For more implementation details, please refer to the Appendix.

**Sets of Hyper-parameters.** We find that several hyper-parameters are often manually tuned, and we chose them to check the robustness of metrics. Totally we will change at most six hyper-parameters of the training method: **1) Early-stopping step** (train-epoch): For the UDA problem, the model at the final step is usually not the best model during training. We divide the total training step into ten epochs and evaluate the model after each epoch. **2) Learning rate** (lr): The initial learning rate **3) Weight decay** (wd) **4) Trade-off**: The trade-off between the supervised loss on the source domain and the target loss from UDA methods. **5) Bottleneck dimension**: The feature dimension output by the feature generator. **6) Hyper-parameter related to training methods**: We choose the margin $\gamma$ (Zhang et al., 2019) for MDD and the temperature $T$ (Jin et al., 2020) for MCC. For DANN and CDAN, we tune the learning rate of the domain discriminator as the hyper-parameter to balance the convergence of the discriminator and the generator (Heusel et al., 2017). We define lr-multi-D as the ratio of the learning rate of the discriminator to the generator.

**Unsupervised Evaluation Metrics.** $\mathcal{A}$-distance(Ben-David et al., 2006), $\mathcal{H}\Delta\mathcal{H}$-divergence or MCD (Ben-David et al., 2010; Saito et al., 2018), MDD (Zhang et al., 2019), DEV (You et al., 2019), Entropy (Grandvalet & Bengio, 2004; Vu et al., 2019), SND (Saito et al., 2021), Mutual Information (Shi & Sha, 2012), ISM (ours), ACM (ours). We implement metrics according to the original papers and modify them to be positively correlated with target accuracy. The implementations are listed in the Appendix.

## 4.2 MAIN RESULTS

In this section, we investigate whether unsupervised evaluation metrics satisfy the "Consistency" principle in Section 3.2. We train the model $\{M_l\}_{l=1}^{n_m}$ using the five UDA methods and the hyper-parameters for the coarse hyper-parameter space in the Tab. 3. For each metric, we report the Pearson correlation ("corr") and the deviation of the best model ("dev"). Tab. 4, Tab. 6, and Tab. 5 show the results of UDA metrics for five training methods on VisDA2017, OfficeHome, and DomainNet. As the results show, it is difficult for previous metrics to represent the target accuracy across all training methods. Some metrics can perform well on the transfer task on one of the datasets but did not perform well on all three, which also shows that testing on partial datasets may lead to biased conclusions. Notably, our proposed ISM is consistent with the target accuracy for most training methods. Our ACM achieves better performance for training methods that align features of two domains, e.g., DANN and CDAN, as it can detect the over-alignment problem.

| Training Method | Source only | | DANN | | CDAN | | MDD | | MCC | | ALL | |
|---|---|---|---|---|---|---|---|---|---|---|---|---|
| Metric | corr | dev | corr | dev | corr | dev | corr | dev | corr | dev | corr | dev |
| $\mathcal{A}$-distance | 0.89 | 1.89 | 0.64 | 0.9 | 0.83 | 0.38 | 0.93 | 0.45 | 0.6 | 9.45 | 0.89 | 4.19 |
| MCD | 0.87 | 1.74 | 0.67 | 6.48 | 0.95 | 0.38 | 0.89 | 0.45 | **0.94** | 5.05 | 0.86 | 13.57 |
| DEV | 0.19 | 1.53 | 0.07 | 3.0 | 0.08 | 1.44 | -0.04 | 12.25 | -0.11 | 5.14 | 0.0 | 2.45 |
| Entropy | 0.45 | 3.43 | 0.65 | 5.83 | 0.79 | 0.49 | 0.83 | 1.09 | 0.75 | 1.37 | 0.71 | 3.55 |
| SND | -0.93 | 11.8 | -0.95 | 20.46 | -0.95 | 11.2 | -0.98 | 39.8 | -0.81 | 24.27 | -0.75 | 23.96 |
| MI | 0.48 | 3.21 | 0.65 | 5.83 | 0.79 | 0.49 | 0.92 | 0.87 | 0.76 | 0.93 | 0.71 | 3.11 |
| ISM | 0.85 | **1.33** | **0.87** | 0.7 | 0.96 | 0.41 | **0.98** | 0.28 | 0.92 | 0.6 | **0.91** | 1.13 |
| ACM | **0.94** | 1.41 | 0.8 | **0.27** | **0.98** | **0.29** | 0.93 | **0.04** | 0.84 | **0.15** | 0.87 | **0.29** |

Table 6: The "corr" and "dev" results are averaged over 12 transfer tasks of DomainNet.

| Method | plane | bcycl | bus | car | house | knife | mcycl | person | plant | sktbrd | train | truck | Avg |
|---|---|---|---|---|---|---|---|---|---|---|---|---|---|
| DANN (default) | 81.7 | 38.7 | 77.8 | 85.8 | 67.2 | 76.7 | 65.5 | 57.9 | 81.3 | 50.4 | 88.5 | 61.0 | 69.4 |
| DANN (searched) | 84.8 | 45.5 | 86.9 | 86.8 | 74.0 | 91.3 | 75.7 | 59.7 | 89.9 | 51.2 | 82.3 | 62.2 | 74.2 |
| Gains ($+\Delta$) | +3.1 | +6.8 | +9.1 | +1.0 | +6.8 | +14.6 | +10.2 | +1.8 | +8.6 | +0.8 | -6.2 | +1.2 | **+4.8** |
| CDAN (default) | 84.8 | 51.5 | 78.8 | 85.1 | 70.0 | 90.8 | 69.7 | 58.6 | 88.2 | 48.5 | 80.2 | 65.3 | 72.6 |
| CDAN (searched) | 86.6 | 47.0 | 82.6 | 85.9 | 75.6 | 87.0 | 78.0 | 63.5 | 88.2 | 55.0 | 79.6 | 76.2 | 75.4 |
| Gains ($+\Delta$) | +1.8 | -4.5 | +3.8 | +0.8 | +5.6 | -3.8 | +8.3 | +4.9 | +0.0 | +6.5 | -0.6 | +10.9 | **+2.8** |
| MDD (default) | 68.9 | 59.5 | 89.7 | 89.5 | 67.8 | 94.4 | 73.7 | 50.2 | 93.4 | 59.0 | 79.5 | 66.2 | 74.3 |
| MDD (searched) | 82.0 | 54.6 | 86.9 | 90.7 | 81.4 | 94.6 | 78.2 | 64.4 | 88.4 | 57.2 | 83.1 | 69.4 | 77.6 |
| Gains ($+\Delta$) | +13.1 | -4.9 | -2.8 | +1.2 | +13.6 | +0.2 | +4.5 | +14.2 | -5.0 | -1.8 | +3.6 | +3.2 | **+3.3** |
| MCC (default) | 85.9 | 71.1 | 77.9 | 87.1 | 80.1 | 82.6 | 58.9 | 58.8 | 90.2 | 55.8 | 80.7 | 75.1 | 75.3 |
| MCC (searched) | 88.5 | 69.3 | 79.2 | 91.0 | 81.7 | 85.0 | 71.0 | 64.3 | 92.8 | 61.1 | 80.0 | 77.2 | 78.4 |
| Gains ($+\Delta$) | +2.6 | -1.8 | +1.3 | +3.9 | +1.6 | +2.4 | +12.1 | +5.5 | +2.6 | +5.3 | -0.7 | +2.1 | **+3.1** |

Table 7: Hyper-parameters found by our metric vs. those manually tuned on VisDA2017.

**Comparison of training methods:** We also investigate the consistency of metrics when comparing different methods. Because in practice, we need to determine the best UDA method for the transfer task. We collected all models trained by all five methods with their metric scores and target accuracy. For each metric, we compute Pearson's correlation and the deviation of the best model, and the results are shown in the "ALL" column. As shown in Tables 4, Table 6, and Table 5, when comparing all training methods, maintaining consistency has become more difficult for most metrics. It is worth noting that our ISM and ACM perform well on all three datasets, with the deviation of the best model ("dev") below 2%. Therefore, we can use the proposed unsupervised metrics to decide the best training method and its hyper-parameters for a dataset.

**Robustness property** We show the robustness property of ISM and ACM in the Appendix.

## 4.3 Unsupervised Hyper-parameter Search

Most UDA methods require manual tuning of hyper-parameters for different datasets. It would be ideal to unsupervised find suitable hyper-parameters automatically. In this section, we show that our ACM can be used for the unsupervised search of hyper-parameters. We will conduct unsupervised hyper-parameter searches for four algorithms: DANN, CDAN, MCC, and MDD. For each UDA training method, we first define its hyper-parameter search space, shown in the dense hyper-parameter space in Tab. 3. We set ACM as the target of the hyper-parameter search. We simply utilized the TPE search algorithm (Bergstra et al., 2011) for 50 trials and Optuna's median pruner (Akiba et al., 2019) to speed up the search. For each transfer task in the dataset, we report the target accuracy of the best model found by ACM. We compare this to the performance of the default hyper-parameters for each method used in the TL-Lib (Jiang et al., 2020). Tab. 7 shows the target accuracy of the model found by our metric and the default model on VisDA. For all four training methods, the hyper-parameters found by us outperform those manually tuned by TL-Lib. Unlike previous supervised tuning, our search process requires no label information on the target domain. Results on Office and DomainNet can be found in the Appendix.

## 5 Conclusion

This paper studies the principles that a robust UDA evaluation metric satisfies. By analyzing the drawbacks of the mutual information metric, we propose Inception Score Metric for UDA (ISM) and Augmentation Consistency Metric (ACM). By conducting extensive experiments, we validate the effectiveness of our metrics in a variety of scenarios. Additionally, our research highlights the potential of evaluation metrics to further the development of AutoML in the UDA.

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
