# OpenReview forum: "A Study of Unsupervised Evaluation Metrics for Practical and Automatic Domain Adaptation"
_ICLR.cc/2024/Conference — ICLR 2024 Conference Withdrawn Submission_

### Official Review · Reviewer_PtWR · 2023-10-23

**Soundness:** 3 good
**Presentation:** 2 fair
**Contribution:** 3 good
**Rating:** 6
**Confidence:** 4

**Summary:**

This paper proposes a new metric for domain adaptation which can be used as a good proxy for the actual target performance but without using target labeled data or validation set. They first explore the use of only source accuracy and mutual information metrics and conclude that neither of them alone could be sufficient. They then propose a metric which is a simple combination of both of these. Further, they also add an augmentation based consistency metric to detect possible over-alignment/negative-alignment of target samples. They show string performance over prior works which propose similar metrics for UDA while also exploring possibility of improving numbers of prior works through valid hyper-parameter tuning.

**Strengths:**

- Unsupervised hyper-parameter searching or model comparison in UDA is a very pertinent and important problem, so the motivation and the idea of the work is well-explained.

- The illustrations presented throughout the paper to support the various claims made are useful and well-presented.

- The prospect of improving performance of existing UDA methods through valid hyper-parameter search without unlabeled target domain is interesting.

**Weaknesses:**

- I could not quite get the argument regarding the proposed metric being "attack-proof" when using a different MLP for metric evaluation compared to classification. Also, the analogy to classical adversarial attacks is also unclear. As shown in Sec 3.2.2, one can also train the auxilliary MLP using the loss in Eq 4 and "cheat" the metric. Perhaps the robustness argument could be made more clearer. Similarly, the Tab.3 in the supplementary is also not that clear to me.

- Similarly, Fig c,d,e in Fig.2 are unclear to me. It suggests that the target alignment hurts if we train using the proxy loss for longer, but how is that indicating robustness or lack of it?

- The argument that for misaligned target samples, the features are more unstable to data augmentation seems only qualitatively supported that too using a single example. To further motivate this, the authors might also present some quantitative evidence (how much percentage of augmented samples are misaligned, etc.)

- While the use of target validation set for choosing hyper-parameters is rightfully argued against, I find the use of target validation set for choosing the right metric equally problematic. For example, several choices in the design of the current metric are based on comparisons with target accuracy, which requires access to target labeled samples. This really hasn't addressed the issue of what to do if we have absolutely no target samples available.

- In fig 4, if I understand correctly, the y-axis in each subfigure shows the normalized metric. But why is the "target accuracy" (blue line) reaching 1.00, when the best possible accuracy on OfficeHome is only around 0.7 (70%)? Does this mean there is a configuration of MCC where we can achieve target accuracy of 1.0?

- Minor: The use of name ISM (Inception Score Metric) might indicate that you are using Inception net somewhere, when you are clearly not. I would suggest clarifying this further in the paper.

**Questions:**

I have several questions regarding the robustness properties, the arguments presented to support it as well as numbers in few of the plots, as summarized above. I would request the authors to answer these, and I would be happy to upgrade my rating.

---

### Official Review · Reviewer_8ogz · 2023-10-25

**Soundness:** 2 fair
**Presentation:** 3 good
**Contribution:** 2 fair
**Rating:** 5
**Confidence:** 3

**Summary:**

This paper studies the practical problems in current unsupervised domain adaptation (UDA) community, e.g., model selection and parameter-tuning. Specifically, though the setting for UDA consists of labeled source data and unlabeled target data for training, the hyper-parameters and optimal model among multiple iterations are usually selected by evaluating the model performance on the target domain, i.e., the testing data with ground-truth labels. To avoid the validity problems on selection and tuning, this paper focuses on unsupervised metrics that can measure the transferability of trained models. Such transferability is considered from the perspectives of dependence, robustness and consistency. By incorporating the metrics inspired by the perspectives above, the authors propose an evaluation method for model selection. The experiments on different datasets show that the selected model has a smaller divergence with the optimal model selected via ground-truth labels.

**Strengths:**

+ The motivation is clear and the organization is easy to follow.
+ An unsupervised evaluation method for practical problems in current UDA is proposed.
+ Experiments are conducted on several datasets and superior results are achieved.

**Weaknesses:**

- Some claims are unrigorous and misleading, which makes this submission less technically sound.
- The technical contributions seem to be limited. Though the motivations are clearly presented, the in-depth analysis and reasoning from methodological and theoretical aspects are insufficient.
- Though several terms, e.g., consistency and robustness, are presented, the formal definition and discussion w.r.t the related areas are insufficient.

**Questions:**

Q1. The formal definitions for the proposed evaluation metrics are necessary, e.g., the three terms in Sec. 3.2. In fact, it seems that there have been many metrics that are similar or equivalent to the motivations in Sec. 3.2., e.g., from the views of consistency and attacks, while the in-depth discussions are not properly provided.

Q2. The claim ‘If we view the MI term as the similarity between two domains, this metric formally follows Ben David’s theory (Ben-David et al., 2006), where the source error and the domain discrepancy bound the target error.’ seems to be inappropriate. As the mutual information used in this submission is defined between the covariate $X$ and predicted label $\hat{Y}$, such a term indeed implies the statistical dependence between label and features, i.e., the discriminability w.r.t pseudo labels on the target domain. Some justifications are appreciated.

Q3. Generally, the essentials of Source Accuracy can also be taken as the discriminability on the source domain. Thus, this term is intuitively equivalent to other metrics, e.g., the mutual information between covariate and ground-truth label which is coherent with the MI term used on the target domain. Besides, there are other works that also consider discriminability as metrics. Thus, some justification on the method and further discussion on the related works are indeed necessary.

Q4. Since the MI term implies the discriminability w.r.t pseudo labels on the target domain, I guess the performance degradation in Fig. 2 can also be caused by the over-fitting on the uncertain pseudo labels. Such a phenomenon also appears when the conditional entropy $H(Y|X)$, e.g., the second term in MI, minimization is adopted. Since conditional entropy minimization on the target domain is widely used in DA literature, it is highly expected to provide additional justifications from the perspectives above.

Q5. The idea for input-level consistency, e.g., ‘We interpret this phenomenon as some target features being pulled to the wrong source class to make the target feature distribution the same as the source.’, seems to be equivalent to the conditional shit or cluster alignment in DA research. Such a problem is extensively studied by pioneer works, which are not properly discussed here.

Q6. Since the main ideas are incorporated from existing literature, some in-depth analysis or theoretical results are highly expected to improve the overall quality of this submission. In current form, it seems that only intuitions and empirical results are provided, which are indeed insufficient from the learning and reasoning aspect.

---

### Official Review · Reviewer_ETkW · 2023-10-29

**Soundness:** 3 good
**Presentation:** 3 good
**Contribution:** 3 good
**Rating:** 6
**Confidence:** 4

**Summary:**

This paper focuses on the design of an evaluation metric to facilitate the identification of optimal trade-off parameters for unsupervised domain adaptation (UDA) methods. The authors propose two metrics, namely Inception Score Metric for UDA (ISM) and Augmentation Consistency Metric (ACM), through empirical analysis. To select the most suitable hyper-parameters in a dense hyper-parameter space, extensive experiments are conducted using various UDA benchmarks and different UDA methods. The results demonstrate the efficacy of the proposed metrics compared to alternative ones.

**Strengths:**

1. Selecting the optimal parameters for unsupervised domain adaptation (UDA) is crucial and poses a challenge in ensuring fair comparisons within the UDA field. Hence, the development of an appropriate evaluation metric for UDA is of importance.
2. The motivation behind this research is clear and logical, and the methods employed are based on insightful analysis.
3. To determine the best hyper-parameters within a dense hyper-parameter space, extensive experiments are conducted on multiple UDA benchmarks using various UDA methods. The results demonstrate the effectiveness of the proposed metrics in comparison to alternative metrics.

**Weaknesses:**

1. Self-training is a prominent and effective approach within the field of unsupervised domain adaptation (UDA). However, the proposed method solely focuses on domain-alignment methods. It is recommended to explore another UDA method based on self-training.
2. The paper exclusively addresses the negative transfer resulting from excessive alignment between two domains. However, it is worth noting that negative transfer can arise from various factors.
3. The authors assert that the metric should be independent of the training method; however, it appears that the methods employed utilize training losses during the evaluation process.
4. Given the remarkable achievements of large-language models (LLM) in transfer learning, I am curious about the performance of the proposed metric on other LLM methods, such as CLIP or Visual Prompt, in the context of domain adaptation or transfer learning.

**Questions:**

Please refer to the weakness part.

---

### Official Review · Reviewer_77mm · 2023-10-30

**Soundness:** 2 fair
**Presentation:** 2 fair
**Contribution:** 2 fair
**Rating:** 3
**Confidence:** 5

**Summary:**

The paper explores the issue of evaluating domain adaptation models in the absence of labels from the target domain. It starts by examining the mutual information metric and identifies three problems with it. To address these issues, the authors introduce two new metrics: the Inception Score Metric (ISM) and the Augmentation Consistency Metric (ACM). Based on mutual information, these metrics consider extra factors such as the accuracy of the source domain and the consistency of target domain augmentation. The authors conduct experiments on closed-set domain adaptation for classification tasks to demonstrate the effectiveness of these new metrics.

**Strengths:**

- Validating unsupervised domain adaptation models without labeled target data is a practical and significant challenge that merits increased research focus.

- The paper offers a thorough examination of related work on the unsupervised validation of domain adaptation models.

- Extensive experiments involving a large-scale hyper-parameter space demonstrate the effectiveness of the proposed validation metrics.

**Weaknesses:**

- The proposed metrics lack soundness.
    - Firstly, both of the proposed evaluation metrics, ISM and ACM, rely on the mutual information metric, which assumes that the label distribution in both domains is class-balanced. Moreover, both metrics highly rely on the source domain accuracy. However, the paper's focus on class-balanced closed-set domain adaptation represents an idealized scenario. In realistic domain adaptation situations, the label distribution in the source domain can be uneven, sometimes following a long-tailed distribution, and a significant label shift often exists between the source and target domains. While both DEV [1] and SND [2] study these realistic domain adaptation scenarios, this paper does not, which may lead to both metrics yielding inaccurate and biased results.
    - Secondly, both ISM and ACM are human-curated without adequate motivation and an ablation study explaining the rationale behind their design. For instance, in the formulation of ISM in Eq. (6), the maximum value of "IS" is "log K," but the addition of a "1/2" term lacks a clear explanation. Additionally, while ISM considers the full mutual information term, i.e., IS, ACM focuses solely on the diversity term without including the entropy term. Moreover, both ISM and ACM are a weighted sum of source validation accuracy and target-domain evaluation (mutual information or consistency evaluation), with equal weights of 0.5-0.5. However, this 0.5-0.5 balance acts as a hyperparameter for these metrics. It is essential to explore alternative weight configurations, such as 0.7-0.3 or 0.2-0.8. In summary, the lack of sufficient theoretical and empirical justification for the metrics' design leaves the motivation and soundness unclear.

- The experiments suffer from a lack of comprehensiveness, primarily due to the absence of important evaluation baselines and UDA methods. Notably, the key aggregation baseline discussed in related work should be included in the comparisons, given its demonstrated superior validation performance, as justified in [3]. Furthermore, an earlier empirical study [4] provides a wealth of validation results using ten evaluation baselines and six types of UDA methods, offering a more comprehensive evaluation than this paper. However, this paper falls short by not including a critical baseline known as "Reverse validation" and limiting the experiments to only two types of UDA methods. In contrast, another concurrent work [5] also explores a broader range of UDA methods in its validation experiments. The paper criticizes previous UDA evaluation methods for their limited selection of UDA methods, and this critique appears to apply to this paper as well.

- The scope of the proposed metrics is confined to closed-set UDA in the context of classification tasks. In contrast, previous entropy-based methods like SND and MI exhibit versatility, as they can readily extend to partial-set, source-free UDA, and even UDA tasks related to segmentation, as supported by [2].


**References**

[1] Towards Accurate Model Selection in Deep Unsupervised Domain Adaptation. ICML 2019

[2] Tune it the Right Way: Unsupervised Validation of Domain Adaptation via Soft Neighborhood Density. ICCV 2021

[3] Addressing Parameter Choice Issues in Unsupervised Domain Adaptation by Aggregation. ICLR 2023 notable-top-5%

[4] Three New Validators and a Large-Scale Benchmark Ranking for Unsupervised Domain Adaptation. arXiv:2208

[5] Can We Evaluate Domain Adaptation Models Without Target-Domain Labels? A Metric for Unsupervised Evaluation of Domain Adaptation. arXiv:2305

**Questions:**

- Primary questions are placed in the section on weaknesses.

- As for a minor presentation suggestion: In the experiments section, all experiments serve to showcase the superior performance of the two metrics, while the critical analysis is placed in the appendix, which may appear unnecessary. I recommend replacing Table 7 with a more detailed presentation of the Robustness property.